

# Aerosol radiative impact during the summer 2019 heatwave produced partly by an inter-continental Saharan dust outbreak – Part 2: Longwave and net dust direct radiative effect

Michaël Sicard[1,2], Carmen Córdoba-Jabonero[3], María-Ángeles López-Cayuela[3], Albert Ansmann[4],
Adolfo Comerón[1], María-Paz Zorzano[5,6], Alejandro Rodríguez-Gómez[1], Constantino Muñoz-Porcar[1]

[1]CommSensLab, Dept. of Signal Theory and Communications, Universitat Politècnica de Catalunya (UPC), 08034-Barcelona,
Spain

[2]Ciències i Tecnologies de l'Espai-Centre de Recerca de l'Aeronàutica i de l'Espai/Institut d'Estudis Espacials de Catalunya
(CTE-CRAE/IEEC), Universitat Politècnica de Catalunya (UPC), 08034-Barcelona, Spain

[3]Instituto Nacional de Técnica Aeroespacial (INTA), Atmospheric Research and Instrumentation Branch, Torrejón de Ardoz,
28850-Madrid, Spain

[4]Leibniz Institute for Tropospheric Research (TROPOS), 04318-Leipzig, Germany

[5]Centro de Astrobiología (CSIC-INTA), Ctra. Ajalvir, km. 4, Torrejón de Ardoz, 28850-Madrid, Spain

[6]School of Geosciences, University of Aberdeen, Aberdeen, AB24 3FX, UK

*Correspondence to*: Michaël Sicard (msicard@tsc.upc.edu)

**Abstract.** This paper is the companion paper of Córdoba-Jabonero et al. (2021). It deals with the estimation of the longwave
(LW) and net dust direct radiative effect (DRE) during the dust episode that occurred between 23 and 30 June, 2019, and
coincided with a mega-heatwave. The analysis is performed at two European sites where polarized-Micro-Pulse Lidars ran
continuously to retrieve the vertical distribution of the dust optical properties: Barcelona, Spain, 23-30 June, and Leipzig,
Germany, 29-30 June. The radiative effect is computed with the GAME radiative transfer model separately for the fine- and
coarse-mode dust. The instantaneous and daily radiative effect and radiative efficiency (DREff) are provided for the fine-
mode, coarse-mode and total dust at the surface, top of the atmosphere (TOA) and in the atmosphere. The fine-mode daily LW
DRE is small (< 6 % of the shortwave (SW) component) which makes the coarse-mode LW DRE the main modulator of the
total dust net DRE. The coarse-mode LW DRE starts exceeding (in absolute values) the SW component in the middle of the
episode which produces positive coarse-mode net DRE at both the surface and TOA. Such an unusual tendency is attributed
to increasing coarse-mode size and surface temperature along the episode. This has the effect of reducing the SW cooling in
Barcelona up to the point of reaching total dust net DRE positive (+0.9 W m$^{-2}$) on one occasion at the surface and quasi-neutral
(-0.6 W m$^{-2}$) at TOA. When adding the LW component, the total dust SW radiative efficiency is reduced by a factor 1.6 at
both surface (on average over the episode, the total dust net DREff is -54.1 W m$^{-2}$ $\tau^{-1}$) and TOA (-37.3 W m$^{-2}$ $\tau^{-1}$). A sensitivity
study performed on the surface temperature and the air temperature in the dust layer, both linked to the heatwave and upon
which the LW DRE strongly depends, shows that the heatwave contributed to reduce the dust net cooling effect at the surface



and that it had nearly no effect at TOA. Its subsequent effect was thus to reduce the heating of the atmosphere produced by the dust particles.

## 1 Introduction

It now makes no doubt that extreme air temperatures have an effect on mortality (Basu and Samet, 2002) and so have mineral dust intrusions (Diaz et al., 2017). Studies of heatwaves and elevated temperatures episodes indicate that the main underlying mechanism leading to death is that stress on the cardiovascular and respiratory systems increases during periods of high air temperature (Kunst et al., 1993). During periods of mineral dust intrusions, the effect on human health is associated to the

biological matter and microorganisms transported in the dust plume that can be harmful to humans (Griffin et al., 2007) and/or to the increase of the particulate matter load at ground level when dust is present (Tobias et al., 2011). When both atmospheric phenomena (heatwave and mineral dust intrusion) occur simultaneously, their effects may cumulate. A place on Earth with a high population density and a high probability that both phenomena occur at the same time is southern Europe because of its proximity to the North African deserts, the largest source of mineral dust on Earth (Prospero et al., 2002).

A recent study from Sousa et al. (2019) quantifies the proportion of Saharan dust intrusions over southwestern Europe since 2000 which were actually accompanied by a heatwave. They find that 39 and 14 % of the Saharan dust intrusions over the western and eastern Iberian Peninsula (IP), respectively, were accompanied by a heatwave. Although dust intrusions are more frequent in the eastern (868 cases) than in the western IP (295 cases), their transport towards the eastern IP is often characterized by an accentuated SW-NE tilt in the mean position of the 580-dam geopotential height thickness, which favors

heatwaves in the central Mediterranean, with little incidence in the eastern IP. In their introduction Sousa et al. (2019) stress "an incomplete description of the relationship between such air masses [Saharan dust] originated in the desertic region of northwestern Africa and heatwave episodes" in the IP.

A recent event of heatwave accompanied with a Saharan dust intrusion occurred in June 2019. The event was outstanding in the sense that temperature records were broken in several places in Europe. The month of June 2019 is still the hottest June

ever recorded at European level (Copernicus, 2021). Average temperatures were more than 2°C above normal and if we consider the 5-day period 25-29 June the temperatures were 6 to 10ºC above normal, with local differences even higher in NE Spain, France and the United Kingdom according to a detailed article issued by the Spanish state meteorological agency (AEMET) on the "June 2019 heatwave in the context of the climate crisis" (AEMET, 2019). Although quite recent, the heatwave of 25-29 June 2019, classified as mega-heatwave by some authors, has already been the subject of several studies.

Xu et al. (2020a; 2021) quantified the role of dynamical and thermodynamical processes in triggering this extreme event and investigated how changes of these processes observed over the last decades may have affected the occurrence probability of such extreme events. Ma et al. (2020) and Vautard et al. (2020) used climate models to quantify the role of human contribution through the anthropogenic climate change. Sousa et al. (2019) studied the relationship between heatwaves and Saharan warm air intrusions in the IP in the long-term context. Finally Córdoba-Jabonero et al. (2021) studied the shortwave (SW) dust direct

radiative effect during 23-30 June 2019 at two sites in Barcelona (BCN), Spain, and Leipzig (LPZ), Germany.

The present paper is the companion paper of Córdoba-Jabonero et al. (2021). It deals with the estimation of the longwave and net dust direct radiative effect during 23-30 June 2019. This second part of the study is motivated by the high temperatures caused by the heatwave that accompanied with the dust event. Indeed, high temperature, clear skies and high insolation imply high land surface temperature and thus strong longwave radiation emission. The paper is organized as follows: the description

and discussion of the dust microphysical and optical properties in the longwave spectral range are exposed in Section 2 (instruments, methodology and radiative effects in the shortwave spectral range can be found in the companion paper); Section 3.1 includes the results in terms of longwave and net dust direct radiative effect and radiative efficiency and Section 3.2



presents a sensitivity study on the relationship between heatwave (by means of surface temperature and air temperature in the dust layer) and coarse-mode longwave and total dust radiative effects. Conclusions are given in Section 4.

## 2 Dust radiative properties in the longwave spectral range and GAME parametrization

The longwave (LW) module of the GAME radiative transfer model (Dubuisson et al., 1996; 2004; 2005) has been used in a recently increasing number of studies (Sicard et al., 2014a; 2014b; Barragan et al., 2016; 2017; 2020; Granados-Muñoz et al., 2019a; 2019b). GAME calculates spectrally integrated, upward and downward radiative fluxes in 40 plane and homogeneous layers from 0 to 100 km with a 1-km resolution from 0 to 25 km. Spectral limits were set in terms of wavenumber from 200 to 2500 cm$^{-1}$ (wavelength: $4 - 50$ μm) at a fixed resolution of 20 cm$^{-1}$ (115 points). GAME accounts for thermal emission, absorption and scattering, as well as their interactions, using the Discrete Ordinates Method (DISORT) (Stamnes et al., 1988). Gaseous absorption ($H_2O$, $CO_2$, $O_3$, $N_2O$, $CO$, $CH_4$ and $N_2$) is treated from the correlated k distribution (Lacis and Oinas, 1991). More details of the longwave module of GAME can be found in Sicard et al. (2014a). GAME presents the advantage of the complete representation of the longwave aerosol scattering, in addition to their absorption. The moderate spectral resolution of GAME makes it possible to account for the spectral variations of aerosol properties, especially in the infrared window. The spectral optical properties of aerosols are defined for each atmospheric layer where dust is present: the single scattering albedo (SSA) and the asymmetry factor (asyF) are assumed constant vertically; the extinction coefficient ($\alpha$) varies with altitude. GAME outgoing (i.e. leaving the terrestrial atmosphere) longwave radiation (OLR) was validated through comparison with CERES (Clouds and the Earth's Radiant Energy System) OLR measurements in 11 cases of dust intrusion in Barcelona (Sicard et al., 2014a). Their results indicate a bias between simulated and measured OLR of -0.8 % and a root mean square error of 2.52 W m$^{-2}$.

The convention followed for the definition of the dust direct radiative effect (DRE) at either the surface (SRF) or the Top-Of-the-Atmosphere (TOA) is the one of Eq. (9) of Córdoba-Jabonero et al. (2021). The atmospheric (ATM) DRE is the difference between the TOA DRE and the SRF DRE. To simplify abbreviations the shortwave, longwave and net (shortwave + longwave) DRE are noted $DRE_{SW}$, $DRE_{LW}$ and $DRE_{NET}$, respectively.

GAME is used to calculate the instantaneous longwave radiative effect in a hourly basis between 5 and 19 UTC over the 8 days of the event in Barcelona (23-30 June) and over 2 days in Leipzig (29 June, Episode 1, and 30 June, Episode 2; see Córdoba-Jabonero et al., 2021). In the rest of this section the dust microphysical and radiative properties are sometimes averaged over the whole event. For Barcelona this means from 23 to 30 June or from 24 to 29 June in order to avoid the beginning and the end of the dust intrusion when abrupt changes in the microphysics and radiative properties are expected. In Leipzig the dust microphysical and radiative properties are averaged over the afternoons of 29 and 30 June (when the dust was present, see Figures 5 and 6 of Córdoba-Jabonero et al., 2021), respectively noted 29J-pm and 30J-pm, corresponding to the two dust episodes described in the former reference. In all cases the averaging period is always indicated.

### 2.1 Dust microphysics

The dust radiative properties in the longwave spectral range were calculated using a Mie code in the range $4 - 50$ μm. The input of our Mie code is the geometric median radius, $r_g$, and its standard deviation, $\sigma_g$, of the lognormal distribution, the particle number, and the spectrally-resolved refractive index. The code calculates the extinction coefficient normalized to that at 532 nm (the wavelength of the P-MPL systems) spectrally-resolved in the range $4 - 50$ μm, $\alpha/\alpha_{532}$, the single scattering albedo and the asymmetry factor. Although the particle number is provided in input, it has no effect on our calculations since SSA and asyF are intensive parameters and the extinction is normalized to that at 532 nm, thus the dependence on the particle





number is removed. We distinguish between the dust coarse-mode (Dc) and the dust fine-mode (Df). The total dust (DD) is the sum of both Dc and Df components.

The spectral refractive index (real and imaginary part) is the same than in Sicard et al. (2014a) and comes from measurements of long range transport mineral dust taken in Meppen in western Germany (Volz, 1983). The table giving the

refractive index as a function of the wavelength was found in Krekov (1993). The spectral variation of both real and imaginary parts of the refractive index can be seen in Figure 1 of Sicard et al. (2014a).

For each of the coarse and fine modes, the geometric median radius and the standard deviation were estimated from column-integrated AERONET retrievals. AERONET provides volume median radius, $r_V$, and its standard deviation, $\sigma_V$. The geometric median radius and standard deviation were calculated as follows:

$$r_g = r_V e^{-3\left(ln\sigma_g\right)^2} \tag{1}$$

$$\sigma_g = \sigma_V \tag{2}$$

AERONET V3L2.0 data were used at both sites. 36 retrievals are available in BCN over the period 23-30 June, and 13 retrievals are available in LPZ for both days of 29 and 30 June. To have an hourly estimation of both $r_g$ and $\sigma_g$ both parameters were interpolated. Both retrievals and hourly interpolated values for $r_V$, $\sigma_V$ and $r_g$ are represented as a function of time in

Figure 1 at both sites and for both size modes. In BCN, the mean (over 23-30 June) Dc and Df geometric median radii (and standard deviation) are 1.262 (0.652) and 0.063 (0.600) μm, respectively; in LPZ we find for the mean of 29J-pm and 30J-pm: $r_g(Dc) = 1.399$ μm ($\sigma_g(Dc) = 0.680$) and $r_g(Df) = 0.032$ μm ($\sigma_g(Df) = 0.505$). Interestingly, the mean large dust particles in LPZ are larger than in BCN and their sizes distribute over a larger scatter interval; $r_g(Dc)$ in LPZ on 29J-pm and 30J-pm (~1.4 μm) is similar to $r_g(Dc)$ in BCN on 29 and 30 June, possibly indicating a conservation of the coarse mode size

distribution during transport between both sites. The small dust particles in LPZ are on average much smaller than in BCN (Figure 1b). It is known that dust aging encompasses many processes that can alter the dust chemical composition, shape and size. In particular, when the transport of mineral dust occurs over polluted regions anthropogenic inorganic acids can be absorbed by the dust surface forming hygroscopic salt compounds that coat the dust particles (Abdelkader et al., 2015; Athanasopoulou et al., 2016). Dust also favors the formation of secondary pollutants (Querol et al., 2019; Xu et al., 2020b).

Both processes (acid absorption and secondary aerosol formation) can lead to dust particle growth in different size ranges. Secondary aerosol formation is enhanced in stagnant (low winds) and high humidity conditions (Xu et al., 2020b). NCEP (National Centers for Environmental Prediction) reanalysis horizontal wind in BCN (not shown) reveals strong winds (~18 m

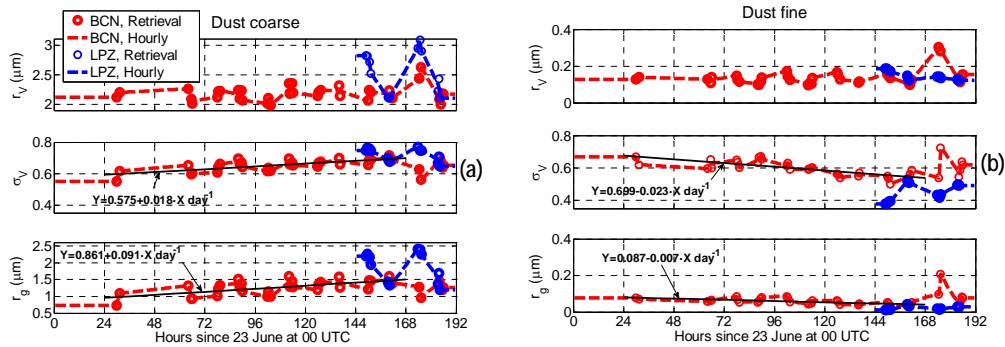

**Figure 1. AERONET volume median radius and standard deviation and corresponding geometric median radius in BCN (red) and**

**LPZ (blue)for (a) the coarse mode and (b) the fine mode. Circles indicate retrievals and the dash lines represent the hourly interpolated data. The legend in the first plot applies to all plots. The black lines are a linear fit of the retrievals in BCN over the period 24-29 June.**




s⁻¹) at the beginning of the episode and much more stagnant conditions (<5 m s⁻¹) towards the end of the episode. All in all, such conditions might have favored the transfer of small dust particles to larger sizes and the growth of coarse-mode particles

along the transport. A linear fit (black lines in Figure 1) has been applied to the BCN retrievals of $r_g$ and $\sigma_g$ over the period 24-29 June in order to avoid the beginning and the end of the dust intrusion when abrupt changes in the microphysics are expected. Coarse and fine modes have opposite tendencies: $r_g(Dc)$ increases at a rate of almost +10 % day⁻¹ and its scatter interval ($\sigma_g$) also increases (~ +2 % day⁻¹); $r_g(Df)$ slightly decreases (at a rate of ~ -1 % day⁻¹) and its scatter interval ($\sigma_g$) also decreases (~ -2 % day⁻¹). The coarse mode radius increase and widening reflects the possible size growth mentioned earlier

that might have occurred during the dust transport. This fact is also nicely illustrated by Figure 2 in which both daily Dc and Df normalized size distributions are represented on a day-by-day basis. One sees clearly how the Dc radii increase between 24 and 29 June, while the Df radii decrease during the same period. Although the decreasing rate of the Df radius along the event is small (< 1 % day⁻¹), its representation in a logarithmic scale (X axis) in Figure 2 shows clearly that it is noteworthy.

**2.2 Dust radiative properties in the longwave spectral range**

With the microphysics defined in the previous section, spectral dust radiative properties ($\alpha$, SSA and asyF) in the longwave spectral range are calculated with the Mie code on a hourly basis at both sites. Results are presented in Figure 3 separately at BCN and LPZ and for Dc and Df. In BCN a color code is used for each curve corresponding to the number of hours past since the beginning of the period considered: 23 June at 00 UTC. In LPZ the average of 29J-pm (Episode 1) and 30J-pm (Episode 2) is shown. The effect of the increasing $r_g(Dc)$ along the dust event in BCN is visible on the Dc extinction plot (Figure 3a,

top panel): $\alpha/\alpha_{532}$ is smaller at the beginning of the event (blue-green curves below the mean) than at the end (red-brown curves above the mean). The main differences between Dc and Df are: the Df extinction is at least three orders of magnitude smaller than the Dc one (in LPZ it is even smaller); Df SSA converges rapidly towards 0 with increasing wavelengths and so does Df asyF. Differences between BCN and LPZ are essentially due to the differences in the dust radii at each site: the mean Dc extinction in BCN is slightly lower than in LPZ ($r_g(Dc)$ is smaller in BCN than in LPZ) and the mean Df extinction in

BCN is larger than in LPZ ($r_g(Df)$ is twice larger in BCN than in LPZ). SSA and asyF are similar at both sites.

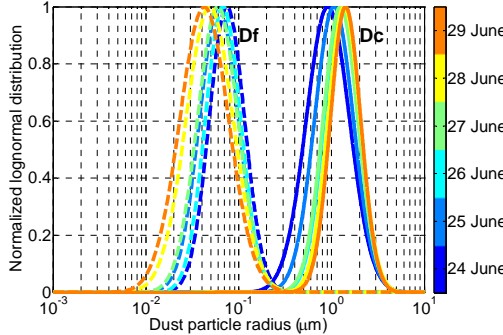

**Figure 2. Day-by-day evolution of the daily mean normalized lognormal distributions in Barcelona between 24 and 29 June calculated with the daily mean ($r_g$, $\sigma_g$) of Dc (solid lines) and Df (dash lines).**

The profiles of extinction coefficients at 532 nm for both coarse and fine modes were taken from the POLIPHON

inversion performed by Córdoba-Jabonero et al. (2021). $\alpha_{532}(Dc)$ and $\alpha_{532}(Df)$ are averaged in 1-km layer-mean values from 0 to 10 km and then input in GAME. In each layer GAME calculates the LW spectral extinction by multiplying the measured $\alpha_{532}$ by the calculated normalized extinction, $\alpha/\alpha_{532}$. Figure 4 represents the Dc and Df dust optical depth (DOD) at 532 nm at both sites. In BCN Dc and Df DOD peak at 0.420 and 0.090, respectively. In LPZ both episodes are much more





moderate ($DOD(Dc)$ peaks at 0.072 and $DOD(Df)$ at 0.045). Except on 24 and 25 June, the Dc DOD variations in BCN are
smooth; for the fine mode DOD swings up and down. In LPZ, DOD outside the dust episodes is very small (< 0.006).

To get an idea of the dust stratification we plot in Figure 5 the dust layer center of mass (CoM) of both size modes and
at both sites. A striking feature is the shape similarity in BCN between Dc DOD (Figure 4a) and Dc CoM (Figure 5a). Such
similarity is usually not expected and probably reflects the homogeneous vertical mixing of the dust plume arriving in BCN
all along the intrusion. In BCN the Dc and Df CoM are quite similar. A peak at ~4 km is observed at the beginning of the event
and next a general temporal decreasing tendency is observed. Towards the end of the intrusion CoM is ~2 km. In LPZ the
centers of mass of the Dc layer are similar to the CoM in BCN at the beginning of the event (23-24 June). Dc CoM of Episode
1 (3.7 − 4.5 km) is higher than that of Episode 2 (3.3 − 3.6 km) as noted by Córdoba-Jabonero et al. (2021). It is noteworthy
to mention that the dust CoM in BCN falls within the climatological dust bottom and top heights ,1.5 and 3.5 km, respectively,

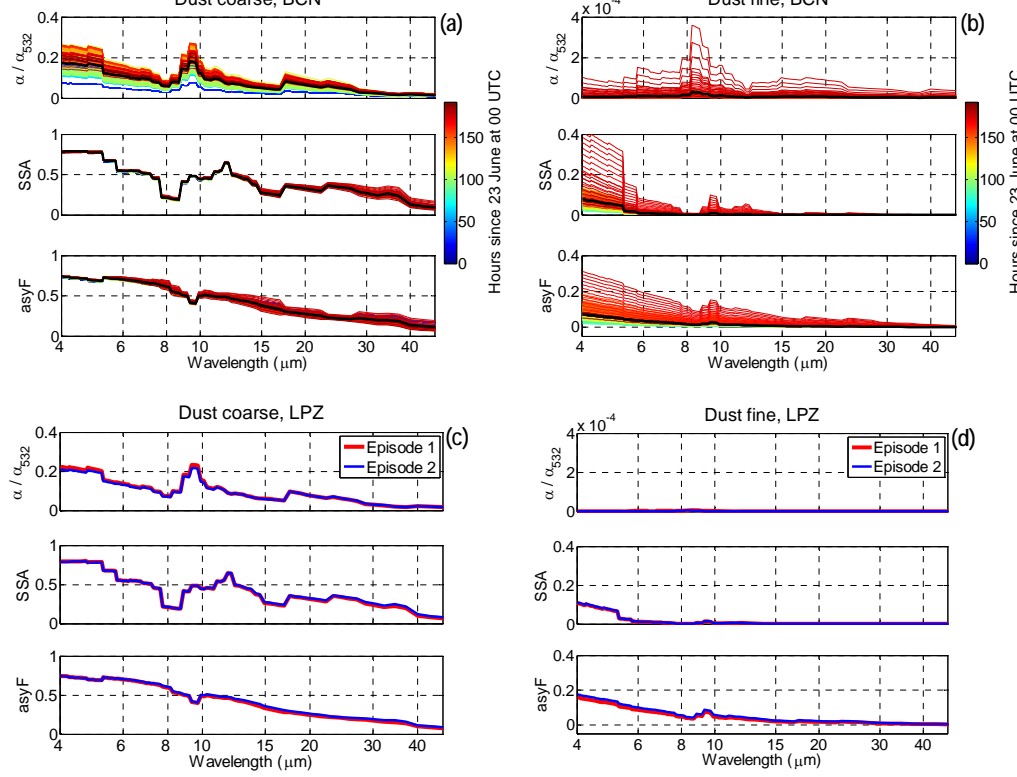

185

**Figure 3. Spectral normalized extinction, α/α$_{532}$, SSA and asyF in BCN for (a) Dc and (b) Df, and in LPZ for (c) Dc and (d) Df. In
BCN all hourly calculations are represented: 192 (8 days). In LPZ the average of 29J-pm (Episode 1) and 30J-pm (Episode 2) is
represented. In the BCN plots the thick black line represents the mean over the whole period (23-30 June).**

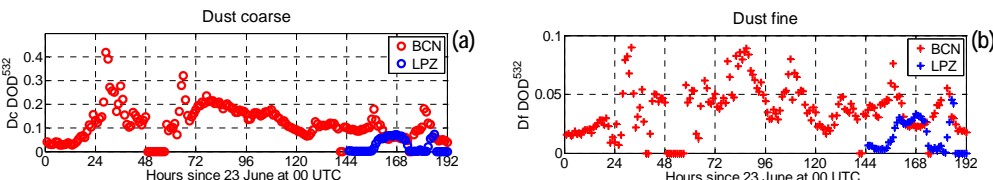

190 **Figure 4. Hourly dust optical depth calculated from the extinction profiles for (a) the coarse mode and (b) the fine mode obtained
using the POLIPHON algorithm in BCN (red) and LPZ (blue).**





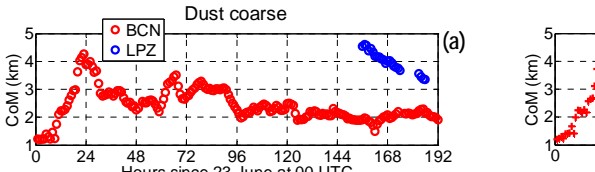

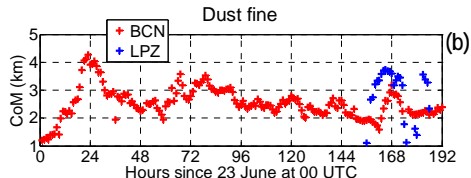

**Figure 5. Hourly dust layer center of mass (CoM) for (a) the coarse mode and (b) the fine mode in BCN (red) and LPZ (blue).**

estimated by Papayannis et al. (2008) from lidar measurements in BCN.

**2.3 GAME parametrization**

Finally the rest of input in GAME are the atmospheric profiles and the Earth surface properties. The gas absorption is parametrized from profiles of pressure, temperature and relative humidity. In BCN, radiosoundings launched twice a day (at 00 and 12 UTC) by the University of Barcelona in collaboration with the Servei Meteorològic de Catalunya, the Catalonia meteorological agency, were used. No radiosoundings are available in LPZ, thus the 6-hour profiles from the Global Data Assimilation System (GDAS) provided by the National Oceanic and Atmospheric Administration (NOAA) were used instead.

The Earth surface is assumed lambertian in GAME and its albedo in the longwave spectral range was set to a constant value of 0.017. This value corresponds to the climatological value found by Sicard et al. (2014a) in Barcelona from CERES measurements in the spectral window 8.1 – 11.8 μm and averaged over spring and summer seasons in the period June 2007 – May 2012. The same value was used at both sites.

We used the hourly land surface temperature (LST) V2 product provided globally at 5-km resolution by the Copernicus Global Land Service (https://land.copernicus.eu/global/products/lst). The LST V2 datasets are estimated from TOA brightness temperatures from the infrared spectral channels of a constellation of geostationary satellites, Meteosat Second Generation being the one covering Europe. Its estimation further depends on the albedo, the vegetation cover and the soil moisture. Further details about the temperature retrieval can be found in Freitas et al., (2013). The LST V2 product user manual and validation reports can be found at https://land.copernicus.eu/global/products/lst?qt-lst_characteristics=5#qt-lst_characteristics. Figure 6 shows the hourly LST in Barcelona (23-30 June) and Leipzig (29-30 June). The diurnal cycle is nicely visible at both sites on all days. In Barcelona the temperature steadily increases between 23 and 28 June. On 29 June the LST starts its decrease. A maximum LST of 45.22 ºC is reached in the afternoon of 28 June. In Leipzig the LST increase is even more pronounced than in Barcelona. The maximum night/day LST difference is 32.2 ºC on 29 June and the maximum LST (46.2 ºC) is reached in the afternoon of 30 June.

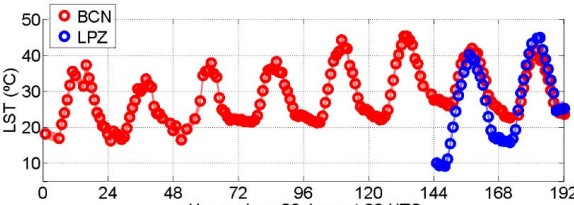

**Figure 6. Hourly LST in BCN (red) and LPZ (blue). The associated uncertainty is added as a shaded area, although it is so small that it is almost imperceptible.**

**3 Results**

The results are first discussed for the longwave spectral range and for the whole spectral range (net effect = shortwave + longwave) in terms of dust direct radiative effect (Section 3.1.1), then in terms of radiative efficiency (Section 3.1.2) and a deeper analysis is performed at BCN site (Section 3.2). All shortwave magnitudes were taken from the Part 1 paper by





Córdoba-Jabonero et al. (2021). Instantaneous (from 5 to 19 UTC) and daily averages are presented. Daily $DRE_{LW}$ is difficult to estimate because GAME calculates longwave fluxes only during daytime hours, so nighttime $DRE_{LW}$ estimations are not available, although they are not zero unlike in the shortwave spectral range. Here the daily $DRE_{LW}$ has been calculated as the

average of the 15 estimations available everyday (from 5 to 19 UTC). Di Sarra et al. (2011) used the same method averaging instantaneous $DRE_{LW}$ obtained every 6 hours. Meloni et al. (2015) assumed their instantaneous $DRE_{LW}$ retrieval constant throughout the 24 hours.

**3.1 Longwave and net dust direct radiative effect**

**3.1.1 Dust direct radiative effect**

The calculations of the instantaneous dust $DRE_{LW}$ are presented as a function of time in both BCN and LPZ in Figure 7 separately for Dc, Df and DD and at SRF, TOA and in the atmosphere. At BCN site we also plot the temporal evolution of the Df/DD $DRE_{LW}$ ratio along the eight days of the event. Daily values of Dc, Df and DD $DRE_{LW}$ are reported at SRF in Table 1 and at TOA in Table 2. Both size modes produce a positive radiative effect at both SRF and TOA. Except on a couple of occasions on 25 June, $DRE_{LW}$ at SRF is always larger than at TOA resulting in a negative atmospheric $DRE_{LW}$, indicator of

a cooling effect of the atmosphere.

Independently of the atmospheric level where it is estimated, the Df $DRE_{LW}$ is small: it is in general more than one order of magnitude smaller than Dc $DRE_{LW}$, and it is smaller at TOA than at SRF. In BCN LW Df instantaneous values do not exceed +0.6 W m⁻² (at SRF, Figure 7) and the daily values are smaller than +0.42 W m⁻² (Table 1). Instantaneous Df $DRE_{LW}$ are in the range of values of Sicard et al. (2014b) also estimated for the dust fine mode: [+0.4; +0.7 W m⁻²] and [+0.1; +0.3 W

m⁻²] at SRF and TOA, respectively. In LPZ instantaneous (daily) Df $DRE_{LW}$ are smaller than +0.2 (+0.08) W m⁻². In BCN the contribution of Df $DRE_{LW}$ to the total dust $DRE_{LW}$ is smaller than 5-6 % (Figure 7c). In average over the period 24-30 June (23 June is discarded because it is the day the dust arrived in BCN) it is 3.6 and 2.5 % at SRF and TOA, respectively, and it seems to slightly decrease with time along the event (negative slope of the fittings in Figure 7c). Hence Df $DRE_{LW}$ is expected to have a very little effect on $DRE_{NET}$ with respect to Dc $DRE_{LW}$. This is also true with respect to Df $DRE_{SW}$ as nicely

illustrated in Figure 8 which shows for Dc, Df and DD the daily LW/SW DRE ratio in absolute value at SRF, TOA and ATM. In the middle plot (Df) the LW/SW DRE ratio is not higher than 10.0 and 5.2 % at SRF and TOA, respectively. The average over the whole event is 6.7 and 3.7 % at SRF and TOA, respectively. Note that the decrease of the Df-to-DD $DRE_{LW}$ ratio with respect to time (opposite of what was observed for $DRE_{SW}$, see Córdoba-Jabonero et al. (2021)) is due to a combination of various factors such as $r_g(Dc)$ (increases), $r_g(Df)$ (decreases), Dc DOD (decreases) and Df DOD (constant).

Unlike the Df component, Dc $DRE_{LW}$ is quite significant. It is also smaller at TOA than at SRF. In BCN Dc instantaneous values reach +11.4 W m⁻² (at SRF) and the daily values are as high as +9.8 W m⁻². In LPZ the instantaneous and daily values (at SRF) do not exceed +3.3 and +1.3 W m⁻², respectively. For comparison, Sicard et al. (2014b) found instantaneous Dc $DRE_{LW}$ in the range [+2.7; +10.2] and [+0.5; +4.7 W m⁻²] at SRF and TOA, respectively. As far as the daily LW/SW Dc DRE ratio is concerned (Figure 8), one sees that at both SRF and TOA it increases with time along the event. It is ~80 % (at both

SRF and TOA) on the first day of the event and it reaches 170 % at SRF (on 28 June) and 183 % at TOA (on 30 June). On the last four days of the event, the daily LW/SW Dc DRE ratio is higher than 119 %, indicating that Dc $DRE_{SW}$ is going to be fully compensated by Dc $DRE_{LW}$, and that, on the daily time scale, the sign of Dc $DRE_{NET}$ will be that of the longwave component, i.e. a positive (warming) DRE. This result is further explored in Section 3.2.

Because Df $DRE_{LW}$ is small compared to Dc $DRE_{LW}$, the total dust $DRE_{LW}$ is similar to that of the coarse mode. In

BCN instantaneous values do not exceed +12.1 W m⁻² at SRF and +7.9 W m⁻² at TOA. The average over the whole event of the daily values are +6.12 W m⁻² at SRF and +3.50 W m⁻² at TOA, respectively, resulting in a LW/SW DD DRE ratio of 67 and 60 %. The fraction of the LW component compared to the SW one is thus similar at both SRF and TOA, and it is higher





than other estimations from the literature. Di Sarra et al. (2011) found daily LW/SW ratios of 49 and 35 % at SRF and TOA, respectively, and Meloni et al. (2015) of 52 and 26 %. Both works deal with desert dust measurements in the Mediterranean, i.e. above the sea. The explanation why our daily LW/SW ratios are higher than those reported in the literature probably lies in the surface temperature. Indeed the LST at solar noon used here is over +15ºC higher than the sea surface temperature considered in Di Sarra et al. (2011) and Meloni et al. (2015), which contributes to enhance $DRE_{LW}$. In LPZ instantaneous values do not exceed +3.3 W m$^{-2}$ at SRF and +3.4 W m$^{-2}$ at TOA. The average over the period 29-30 June of the daily values are +0.96 W m$^{-2}$ at SRF and +1.04 W m$^{-2}$ at TOA, respectively. It is interesting to observe that in LPZ $DRE_{LW}$ is higher at TOA than at SRF (in BCN the reverse occurs). As shown by Dufresne et al. (2002) and Sicard et al. (2014a) $DRE_{LW}$ is highly dependent on the dust layer height and its variations with the latter at SRF and TOA are opposite ($DRE_{LW}$ decreases at SRF with increasing height while it increases at TOA). The height of the coarse dust layer is much higher in LPZ than in BCN (Figure 5a), which enhances $DRE_{LW}$ at TOA with respect to the surface. The DD $DRE_{LW}$ does not follow systematically a constant diurnal cycle like the SW one does. On the first three days no pattern is identified, probably because of the strong variations of the DOD (see Figure 4). On 26 June DD $DRE_{LW}$ is rather flat, while on the last four days of the event both SRF and TOA $DRE_{LW}$ have a clear diurnal cycle with the shape of a semi cosine curve (28 June) or an inverted-V (27, 29 and 30 June) showing an increase in the morning, a maximum reach at 12 or 13 UTC and a decrease afterward. This pattern of the total dust $DRE_{LW}$ is mostly due to the coarse mode since the fine mode $DRE_{LW}$ is much smaller than the coarse one. The diurnal cycle of DD (or Dc) $DRE_{LW}$ is more pronounced at TOA than at SRF. It is well known that the OLR has a marked diurnal cycle (Slingo et al., 1987) and that it is highly correlated to the surface heating and cooling due to the diurnal cycle of insolation (Chung et al., 2009). For $DRE_{LW}$ these statements are less straightforward. When holding all parameters constant except the surface temperature, Osborne et al. (2011) showed with Saharan dust measurements performed in Mauritania and Niger in June 2007 that the shape of the diurnal cycle of $DRE_{LW}$ mimicked that of the surface temperature. In our analysis the surface temperature is also the only variable with a diurnal cycle with the shape of a cosine curve, so we believe that the cosine or inverted-V shape of DD $DRE_{LW}$ observed on the last four days of the dust event is related to the diurnal cycle of the surface temperature.

The instantaneous dust $DRE_{NET}$ ($DRE_{SW} + DRE_{LW}$) is presented as a function of time in both BCN and LPZ in Figure 9 separately for Dc, Df and DD and at SRF, TOA and in the atmosphere. Daily values of Dc, Df and DD $DRE_{NET}$ are reported at SRF in Table 1 and at TOA in Table 2. The fine mode $DRE_{NET}$ is similar to $DRE_{SW}$ since $DRE_{LW} \ll DRE_{SW}$. It produces a negative net radiative effect at both SRF and TOA. The daily mean values (averaged over the whole event) are -3.2 (BCN) and -1.15 (LPZ) W m$^{-2}$ at SRF and -2.5 (BCN) and -0.85 (LPZ) W m$^{-2}$ at TOA. Dc and DD instantaneous $DRE_{NET}$ at SRF and TOA are similar to their corresponding $DRE_{SW}$ but shifted towards positive values (Figure 9). At SRF daily Dc $DRE_{NET}$ is negative in the first half period of the event in BCN and in LPZ. In BCN the average over the whole event is +0.2 W m$^{-2}$, i.e. that at SRF the LW component wins over the shortwave one at the event scale. At TOA daily Dc $DRE_{NET}$ is essentially positive in BCN (except on the first three days of the event) and in LPZ. In BCN (LPZ) the average over the whole event is +0.2 (+0.5) W m$^{-2}$, i.e. that at TOA the LW component wins over the shortwave one at the event scale and at both sites. It makes no doubt that for this particular dust episode and for the coarse-mode dust particles the longwave DRE is as important as the shortwave one. Not having it taken into account would have led to large errors in the final estimation of the total dust $DRE$ and to an overestimation of the dust cooling effect. Because Dc $DRE_{SW}$ and $DRE_{LW}$ nearly compensate, the dust coarse mode net contribution to the total dust $DRE_{NET}$ is overall small (Dc $|DRE_{NET}| < 0.5$ W m$^{-2}$), independently of the atmospheric level. Finally, as far as DD $DRE_{NET}$ is concerned, most of the DD $DRE_{NET}$ instantaneous values at SRF and TOA are negative and remain lower than 20 W m$^{-2}$ in absolute value. The daily means are -3.0 (BCN) and -1.40 (LPZ) W m$^{-2}$ at SRF and -2.3 (BCN) and -0.35 (LPZ) W m$^{-2}$ at TOA. At SRF the highest daily DD $DRE_{NET}$ are -7.4 and -1.5 W m$^{-2}$ in BCN and LPZ, respectively. In one occasion in BCN (on 28 June) the daily DD $DRE_{NET}$ at SRF takes a positive value +0.9 W m$^{-2}$, meaning that on a daily basis the dust radiative effect produces a net warming at SRF. This is quite an unusual results. Our values are





significantly smaller (in absolute value) than the limited number of dust, daily $DRE_{NET}$ that can be found in the literature. For instance, Di Sarra et al. (2011) found daily DD $DRE_{NET}$ of -43.7 (SRF) and -37.6 W m$^{-2}$ (TOA); Meloni et al. (2015) of -14.7 (SRF) and -15.5 W m$^{-2}$ (TOA); and Valenzuela et al. (2017) of [-64; -37 W m$^{-2}$] (SRF) and [-31; -18 W m$^{-2}$] (TOA). As said earlier, the most probable explanation why our values are smaller than the literature lies in the surface temperature which is

much higher in our work.

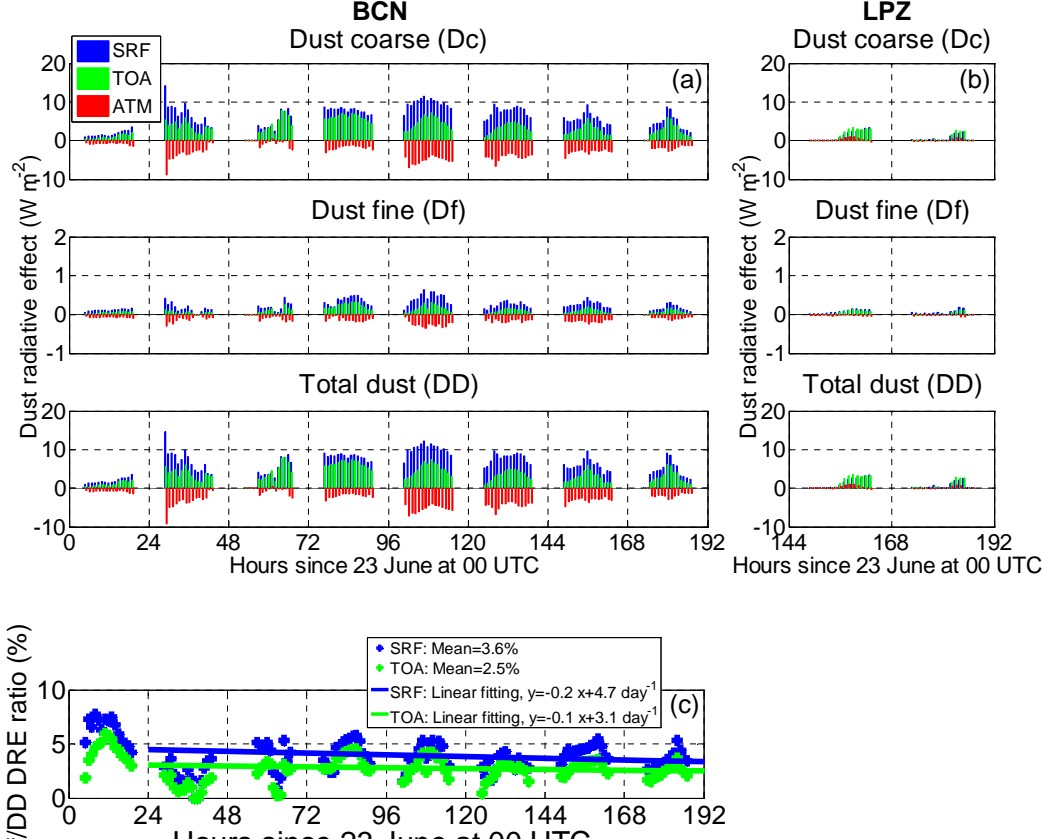

**Figure 7.** Longwave instantaneous dust direct radiative effect ($DRE_{LW}$) at SRF, TOA and in the atmosphere (ATM) in (a) BCN and
**(b) LPZ. (c) Df/DD DRE ratio of the instantaneous $DRE_{LW}$ at the SRF and TOA in BCN; the mean values and best linear fits have
been calculated between 24 and 30 June. The absolute decrease of Df/DD $DRE_{LW}$ ratio at the SRF (TOA) is -0.2 (-0.1) % day$^{-1}$.**

The objective of this last paragraph is to complete for the net radiative effect the discussion about the diurnal cycle of $DRE_{SW}$ made in the companion paper by Córdoba-Jabonero et al. (2021). Figure 10 shows the diurnal cycle of $DRE_{NET}$ for Dc, Df and DD at SRF, TOA and in the atmosphere for 26 June. Dc and DD $DRE_{SW}$ are also reproduced. Df $DRE_{SW}$ is not
reproduced since it is nearly equal to Df $DRE_{NET}$ (Df $DRE_{LW}$ is small). We find again that the diurnal cycle of the Dc and DD $DRE_{NET}$ is singular at SRF and TOA with the shape of a "W", showing two minima, one in the morning (around 6 UTC) and one in the afternoon (17-18 UTC), and a maximum at central hours of the day. This shape is the one of the diurnal cycle of $DRE_{SW}$ (the diurnal cycle of $DRE_{LW}$ is rather flat on 26 June) which is due to a combination of solar geometry and dust anisotropic scattering (Osborne et al., 2011; Osipov et al., 2015). We refer to Córdoba-Jabonero et al. (2021) for a detailed
explanation of the "W" shape of the diurnal cycle of $DRE_{SW}$. At SRF and TOA both Dc and DD $DRE_{NET}$ are similar to $DRE_{SW}$ shifted towards positive values (because of the Dc LW offset). The cooling effect produced during the day by $DRE_{SW}$





Table 1: Daily $DRE_{LW}$ and $DRE_{NET}$ (W m$^{-2}$) and DREff (W m$^{-2}$ $\tau^{-1}$) at SRF produced by Dc, Df and DD particles in BCN and LPZ. $\overline{X}$ indicates daily-averages. The daily DOD at 532 nm, $DOD^{532}$, is also included. Df and DD $DRE_{LW}$ and the 29-30 June mean DRE in LPZ are given with a precision of 2 digits.

| | | BCN | | | | | | | | | LPZ (*) | | |
|---|---|---|---|---|---|---|---|---|---|---|---|---|---|
| June 2019 | | 23 | 24 | 25 | 26 | 27 | 28 | 29 | 30 | 23-30 | 29 | 30 | 29-30 |
| DOD$^{532}$ | $\overline{D_c}$ | 0.055 | 0.189 | 0.098 | 0.193 | 0.140 | 0.090 | 0.092 | 0.071 | 0.116 | 0.028 | 0.020 | 0.024 |
| | $\overline{D_f}$ | 0.019 | 0.041 | 0.031 | 0.062 | 0.045 | 0.029 | 0.039 | 0.031 | 0.037 | 0.016 | 0.013 | 0.015 |
| | $\overline{DD}$ | 0.074 | 0.230 | 0.129 | 0.255 | 0.185 | 0.119 | 0.131 | 0.102 | 0.153 | 0.044 | 0.033 | 0.039 |
| DRE | LW $\overline{D_c}$ | +1.8 | +6.9 | +3.7 | +7.8 | +9.8 | +7.5 | +5.6 | +4.0 | +5.9 | +1.3 | +0.5 | +0.90 |
| | LW $\overline{D_f}$ | +0.11 | +0.16 | +0.14 | +0.35 | +0.42 | +0.24 | +0.26 | +0.14 | +0.22 | +0.08 | +0.05 | +0.06 |
| | LW $\overline{DD}$ | +1.91 | +7.06 | +3.84 | +8.15 | +10.22 | +7.74 | +5.86 | +4.14 | +6.12 | +1.38 | +0.55 | +0.96 |
| | NET $\overline{D_c}$ | -0.2 | -2.7 | -2.0 | -1.7 | +3.0 | +3.1 | +1.2 | +0.6 | +0.2 | -0.2 | -0.3 | -0.25 |
| | NET $\overline{D_f}$ | -1.6 | -3.6 | -3.0 | -5.7 | -3.7 | -2.2 | -3.1 | -2.6 | -3.2 | -1.3 | -1.0 | -1.15 |
| | NET $\overline{DD}$ | -1.8 | -6.3 | -5.0 | -7.4 | -0.7 | +0.9 | -1.9 | -2.0 | -3.0 | -1.5 | -1.3 | -1.40 |
| DREff | LW $\overline{D_c}$ | +39.6 | +34.6 | +30.5 | +40.2 | +64.8 | +71.6 | +54.6 | +48.7 | +44.3 | +46.2 | +32.8 | +41.3 |
| | LW $\overline{D_f}$ | +5.0 | +3.9 | +3.7 | +5.0 | +8.1 | +7.1 | +5.8 | +3.8 | +5.3 | +4.7 | +4.1 | +4.4 |
| | LW $\overline{DD}$ | +28.5 | +29.0 | +23.9 | +30.7 | +50.2 | +55.6 | +39.8 | +34.8 | +34.9 | +37.1 | +26.0 | +28.2 |
| | NET $\overline{D_c}$ | -33.8 | -40.6 | -55.4 | -38.3 | -7.7 | +5.1 | -12.2 | -13.5 | -30.8 | -39.9 | -62.8 | -48.2 |
| | NET $\overline{D_f}$ | -118.0 | -141.6 | -140.2 | -128.5 | -116.7 | -103.9 | -113.3 | -108.5 | -124.4 | -144.1 | -163.5 | -153.5 |
| | NET $\overline{DD}$ | -60.8 | -58.9 | -76.3 | -62.7 | -36.0 | -21.9 | -42.8 | -42.9 | -54.1 | -74.0 | -101.8 | -85.3 |

(*) 29 and 30 June represent the dust Episode 1 and Episode 2, respectively, as observed in LPZ.

Table 2: Same as Table 1 but at the TOA. $DOD^{532}$ values are shown in Table 1.

| | | BCN | | | | | | | | | LPZ (*) | | |
|---|---|---|---|---|---|---|---|---|---|---|---|---|---|
| June 2019 | | 23 | 24 | 25 | 26 | 27 | 28 | 29 | 30 | 23-30 | 29 | 30 | 29-30 |
| DRE | LW $\overline{D_c}$ | +1.0 | +3.7 | +3.2 | +6.1 | +4.6 | +3.2 | +2.7 | +2.6 | +3.4 | +1.4 | +0.6 | +1.0 |
| | LW $\overline{D_f}$ | +0.05 | +0.05 | +0.06 | +0.22 | +0.16 | +0.10 | +0.08 | +0.06 | +0.10 | +0.05 | +0.03 | +0.04 |
| | LW $\overline{DD}$ | +1.05 | +3.75 | +3.26 | +6.32 | +4.76 | +3.30 | +2.78 | +2.66 | +3.50 | +1.45 | +0.63 | +1.04 |
| | NET $\overline{D_c}$ | -0.3 | -2.2 | -0.3 | +0.3 | +0.9 | +1.0 | +0.6 | +1.1 | +0.2 | +0.7 | +0.3 | +0.50 |
| | NET $\overline{D_f}$ | -1.3 | -3.2 | -2.4 | -4.6 | -2.9 | -1.6 | -2.3 | -1.7 | -2.5 | -1.0 | -0.7 | -0.85 |
| | NET $\overline{DD}$ | -1.6 | -5.4 | -2.7 | -4.3 | -2.0 | -0.6 | -1.7 | -0.6 | -2.3 | -0.3 | -0.4 | -0.35 |
| DREff | LW $\overline{D_c}$ | +23.4 | +17.7 | +26.9 | +30.9 | +31.2 | +31.8 | +28.4 | +31.3 | +26.5 | +52.9 | +40.7 | +48.5 |
| | LW $\overline{D_f}$ | +2.0 | +1.1 | +1.9 | +3.2 | +3.2 | +2.7 | +2.1 | +1.8 | +2.4 | +3.7 | +2.8 | +3.3 |
| | LW $\overline{DD}$ | +16.5 | +14.6 | +20.7 | +23.5 | +24.2 | +24.8 | +20.4 | +22.2 | +20.8 | +37.1 | +26.0 | +32.6 |
| | NET $\overline{D_c}$ | -29.9 | -32.3 | -29.8 | -15.6 | -7.4 | -0.9 | +0.1 | +10.8 | -17.4 | +1.0 | -5.7 | -1.4 |
| | NET $\overline{D_f}$ | -95.7 | -133.1 | -117.7 | -99.9 | -83.1 | -73.1 | -76.7 | -66.8 | -96.2 | -108.2 | -119.5 | -113.6 |
| | NET $\overline{DD}$ | -51.3 | -49.8 | -52.1 | -38.6 | -26.9 | -18.8 | -23.1 | -13.2 | -37.3 | -106.3 | -125.6 | -114.2 |

(*) 29 and 30 June represent the dust Episode 1 and Episode 2, respectively, as observed in LPZ.





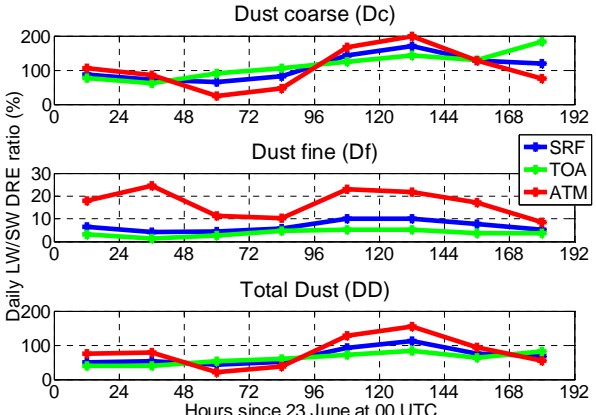

**Figure 8. Daily LW/SW DRE ratio in BCN at the SRF, TOA and in the atmosphere (ATM) for (top) Dc (middle) Df and (bottom) DD.**

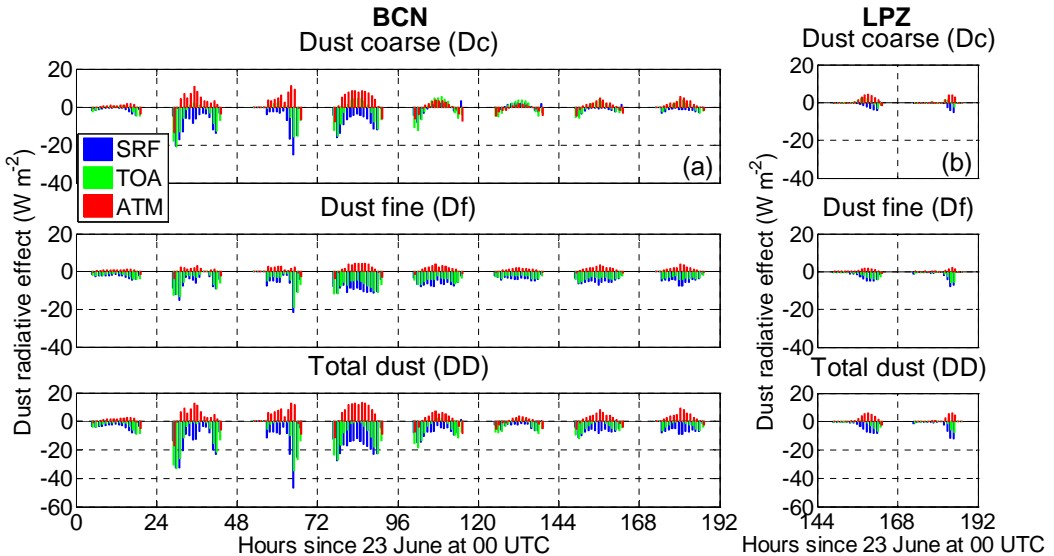

**Figure 9. Net instantaneous dust direct radiative effect ($DRE_{NET}$) at SRF, TOA and in the atmosphere (ATM) in (a) BCN and (b) LPZ.**

is thus partly counterbalanced by $DRE_{LW}$. At TOA during the central hours (11-13 UTC), when Dc $DRE_{SW}$ reaches its minimum, DD even produces a quasi-neutral effect (-0.7 < DD $DRE_{NET}$ < -0.2 W m$^{-2}$). At SRF, the instantaneous DD $DRE_{NET}$ is negative and it is larger (in absolute value) than 12 W m$^{-2}$; the daily DD $DRE_{NET}$ (Table 1) is equal to -7.4 W m$^{-2}$. As expected, when considering a complete daily cycle (24 hours) instead of instantaneous values, the LW DRE plays a larger role with respect to the SW DRE because the latter is zero during the night hours. The diurnal DRE variations at TOA are larger than at SRF. As a consequence, $DRE_{NET}$ in the atmosphere shows the shape of an inverted U for both Dc and DD. The dust (all modes) produces a heating of the atmosphere during most of the hours of the day ($DRE_{NET}$ at TOA > $DRE_{NET}$ at SRF) and a slight cooling at dawn/dusk. The effect of $DRE_{LW}$ on the atmospheric $DRE_{NET}$ is rather small: the difference for Dc and DD between the atmospheric $DRE_{SW}$ and $DRE_{NET}$ is not greater than 2 W m$^{-2}$.





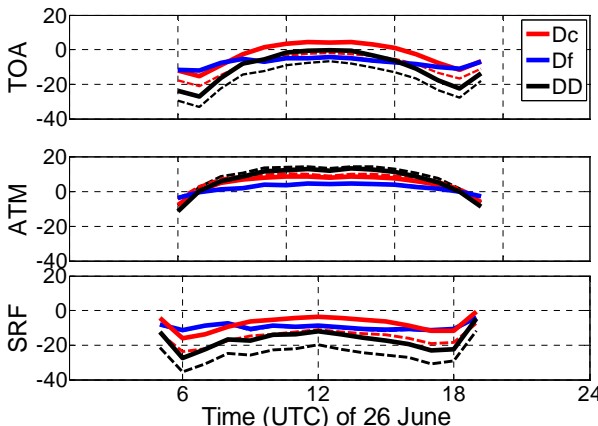

**Figure 10. Diurnal cycle on 26 June in BCN of the dust net radiative effect ($DRE_{NET}$) for Dc, Df and DD (bottom) at SRF, (middle) in the atmosphere, and (top) at TOA. For Dc (red) and DD (black) the diurnal cycle of $DRE_{SW}$ is also reported by dash lines. Df $DRE_{SW}$ is not represented since it would overlap with Df $DRE_{NET}$.**

### 3.1.2 Dust direct radiative efficiency

The dust direct radiative efficiency (DREff) is assessed on a daily basis (and at the event scale) by calculating the best linear
fit forced to 0 of the scatterplot of instantaneous values of one full day (and of the whole dust episode) of $DRE$ vs. $DOD^{532}$.
First the longwave dust direct radiative efficiency is discussed at SRF and TOA: see Figure 11 showing all instantaneous
$DRE_{LW}$ vs. $DOD^{532}$ at SRF and TOA separately for Dc/Df and BCN/LPZ, and Table 1 and Table 2 reporting the daily LW
DREff values; and then the net direct radiative efficiency is discussed compared to the SW and LW components separately for
Dc, Df and DD and at SRF, TOA and in the atmosphere: see Figure 12 showing only the linear fits (for the sake of clarity) of
SW, LW and net DRE components vs. $DOD^{532}$, and Table 1 and Table 2 reporting the daily net DREff values. As expected
Df LW DREff is small at SRF and TOA and at both sites (Df LW DREff < +5.3 W m$^{-2}$ $\tau^{-1}$). At the event scale at SRF Dc LW
DREff in BCN (+44.3 W m$^{-2}$ $\tau^{-1}$) and in LPZ (+41.3 W m$^{-2}$ $\tau^{-1}$) are similar, while at TOA they differ significantly (+26.5 and
+48.5 W m$^{-2}$ $\tau^{-1}$, respectively). There are two reasons for that difference: 1) in average the dust coarse particles are larger in
LPZ than in BCN (see Section 2.1 and Figure 1) inducing larger extinction coefficients in the longwave spectral range (Figure
3a and c) and 2) the center of mass of the dust plume is much higher in LPZ than in BCN (Figure 5) which produces higher
LW DRE at TOA (Dufresne et al., 2002; Sicard et al., 2014a). According to Sicard et al. (2014a), when maintaining the aerosol
optical depth constant, Dc $DRE_{LW}$ does not vary much for $r_g(Dc) > 1$ μm. This result lets us think that the main reason why
Dc LW DREff at TOA in LPZ is higher than in BCN is related with the height of the dust plume. When Df and Dc are
considered together, DD LW DREff drops down (with respect to the Dc value) to +34.9 (BCN) +28.2 W m$^{-2}$ $\tau^{-1}$ (LPZ) at SRF
and to +20.8 (BCN) and +32.6 W m$^{-2}$ $\tau^{-1}$ (LPZ) at TOA.

To study the effects of each particle size mode (Dc and Df) and spectral components (SW and LW) we now look at Figure
12 which has been produced with the data from BCN for the whole event (23-30 June). The central plots show DRE vs.
$DOD^{532}$ for the dust fine mode. Df LW DREff is small at all atmospheric levels which leads to similar values of Df SW and
net DREff (black and red lines almost overlap in Figure 12b, e and h). This allows to conclude that the dust Df net DREff is
driven basically by the SW component. Things are different for the dust coarse mode (left plots in Figure 12) for which the
LW component offsets significantly the SW one. Indeed, at SRF and TOA, we observe a decrease of the net Dc radiative
efficiency with respect to the SW one of a factor 2.5: -30.8 (net) vs. -75.2 W m$^{-2}$ $\tau^{-1}$ (SW) at SRF, and -17.4 (net) vs. -43.9 W
m$^{-2}$ $\tau^{-1}$ (SW) at TOA. In the atmosphere (Figure 12g), the inclusion of the LW component reduces the Dc net DREff of a factor





1.6 with respect to the SW one. When both size modes are added together (DD, right plots of Figure 12), a reduction of the

SW radiative efficiency is also observed when the LW component is taken into account. At SRF and TOA, we observe a decrease of the net DD radiative efficiency with respect to the SW one of a factor 1.6: -54.1 (net) vs. -88.9 W m$^{-2}$ $\tau^{-1}$ (SW) at SRF, and -37.3 (net) vs. -58.0 W m$^{-2}$ $\tau^{-1}$ (SW) at TOA. Interestingly, the ratio of the total dust SW-to-NET radiative efficiency is the same (1.6) at both SRF and TOA. It is a value significantly larger than 1 which highlights again the importance of considering the longwave component in studies focused on dust radiative effects. Finally, in the atmosphere (Figure 12i), the

inclusion of the LW component reduces the DD net DREff of a factor 1.8 with respect to the SW one.

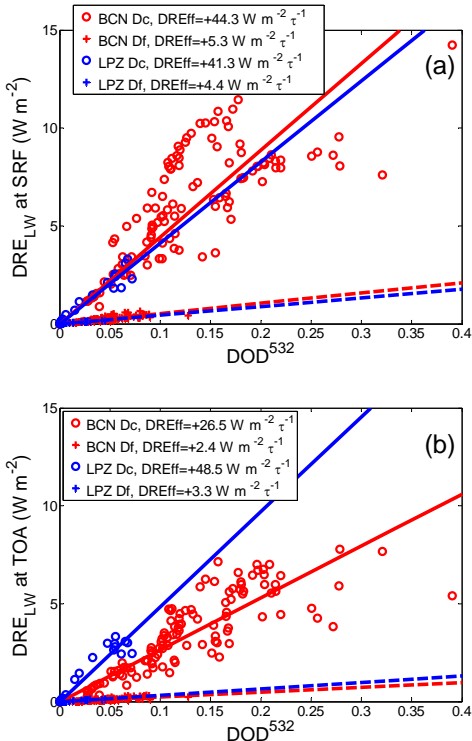

**Figure 11. Dust longwave direct radiative effect ($DRE_{LW}$) at (a) SRF and (b) TOA as a function of $DOD^{532}$, as shown separately for**

**Dc (circles) and Df (crosses) at BCN (23-30 June) and LPZ (29-30 June). Corresponding DREff values (slope of the linear fitting of $DRE_{LW}$ vs. $DOD^{532}$) are included in the legend (solid and dashed lines, respectively, for Dc and Df particles).**

### 3.2 Heatwave and dust cooling/warming effect of the Earth-Atmosphere system

How the net direct radiative effect of mineral dust is modified when the dust intrusion occurs simultaneously with a heatwave? This question is the motivation of this section which focuses on the results in BCN. We will start with some features of the

June 2019 heatwave. Data provided by the Copernicus Climate Change Service show that the European-average temperature for June 2019 was higher than for any other month of June on record (Copernicus, 2021). Average temperatures were more than 2°C above normal and if we consider the 5-day period 25-29 June the temperatures were 6 to 10ºC above normal, with local differences even higher (up to 18ºC!) in NE Spain, France and the United Kingdom according to a detailed article issued by the Spanish state meteorological agency (AEMET) on the "June 2019 heatwave in the context of the climate crisis"

(AEMET, 2019). In June 2019, the synoptic conditions were marked by the presence of an anomalous long-lasting anticyclone in the upper troposphere which advected warm air from the Sahel and Mediterranean region and enhanced incoming solar radiation and surface turbulent fluxes. According to Xu et al. (2020a) this situation is at the origin of three heatwaves over



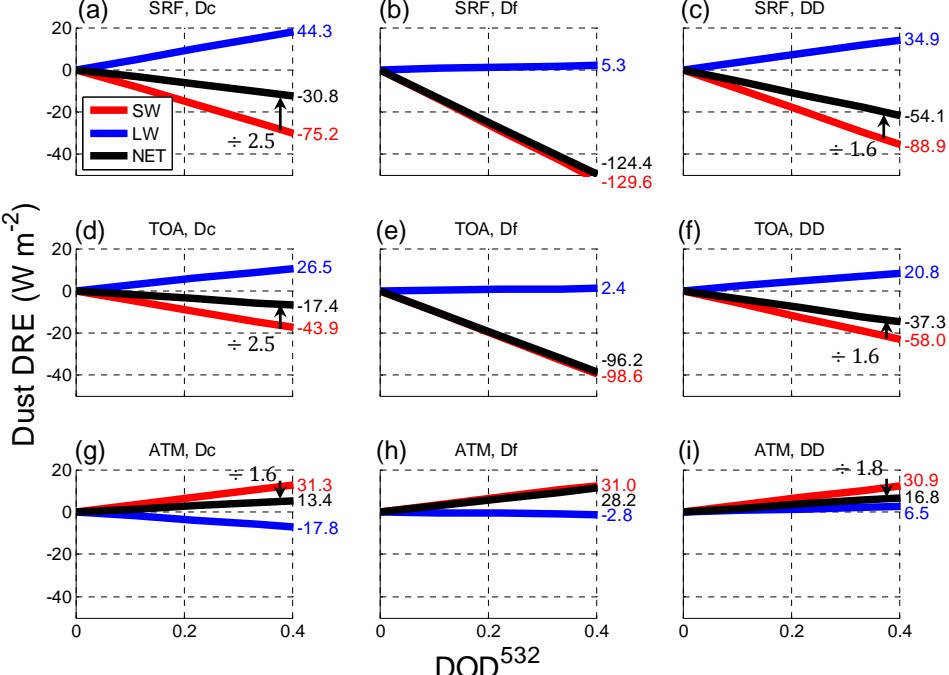

**Figure 12.** Linear fitting of the dust direct radiative effect (SW, LW and net) as a function of $DOD^{532}$ at SRF for (a) Dc, (b) Df and (c) DD; at TOA for (d) Dc, (e) Df and (f) DD; and in the atmosphere for (g) Dc, (h) Df and (i) DD), in BCN (23-30 June). Corresponding DREff values (slope of the linear fitting of DRE vs. $DOD^{532}$) are included next to the right axis. DREff units are W m$^{-2}$ $\tau^{-1}$. In Dc and DD plots the arrows and number indicate the SW / NET ratio. The legend in plot (a) applies to all plots.

Europe and it is during the last one, from 25 to 29 June, that the highest temperatures were reached. Towards the end of June, this anticyclone merged with a high-pressure system located over northeastern Europe that had previously produced extreme temperatures, and extended towards the northeast as a strong subtropical ridge intensified by a low-pressure system located over the eastern Atlantic. This subtropical ridge injected very warm air and mineral dust from the Sahara region towards western Europe, and this air overheated during its transport while traveling over previously warmed land (Sousa et al., 2019; Xu et al., 2020a). Sousa et al. (2019) classified the heatwave of 25-29 June 2019 as a mega-heatwave (Barriopedro et al., 2011) because of its outstanding duration, intensity and spatial extent. Other climate studies involving multi-model methodologies have come to the conclusion that the June 2019 heatwave had been most probably triggered by anthropogenic climate change, and that such heatwaves could become more widespread, long-lasting, and severe over Europe in the future (Ma et al., 2020; Vautard et al., 2020).

As seen in the previous section, the total dust net radiative effect is modulated by the dust coarse mode LW radiative effect. Figure 13a shows on the same plot the daily Dc LW/SW DRE ratio and the daily Df and DD $DRE_{NET}$ at SRF for the BCN site and for the period 23-30 June; Figure 13b shows the same magnitudes at TOA. In Figure 13c the daily LST is reported. A clear correlation exists between the LST and the Dc LW/SW DRE ratio at both SRF and TOA. In the same way a clear correlation exists between Dc LW/SW DRE ratio and DD $DRE_{NET}$ at SRF and also at TOA. 26 June is a pivotal moment of the episode in terms of dust radiative effect: during that day the TOA (first) and SRF (second) Dc LW/SW DRE ratio cross the 100 % threshold (Dc $|DRE_{SW}|$ = Dc $|DRE_{LW}|$). This affects directly DD $DRE_{NET}$ with respect to Df $DRE_{NET}$: DD





$|DRE_{NET}| > $ Df $|DRE_{NET}|$ before 26 June (Dc $DRE_{NET}$ amplifies the dust fine mode cooling effect) and DD $|DRE_{NET}| < $ Df $|DRE_{NET}|$ after 26 June (Dc $DRE_{NET}$ reduces the dust fine mode cooling effect). As expected the difference between Df and DD $DRE_{NET}$ is larger at SRF than TOA which reflects that the contribution of Dc $DRE_{NET}$ is more significant at SRF, where most of the LW radiation comes from, than at TOA. When Dc LW/SW DRE ratio reaches values above 150 % at SRF, DD

$DRE_{NET}$ takes values close to zero (-0.7 and +0.9 W m$^{-2}$ on 27 and 28 June, respectively), i.e. the total dust net radiative effect is quasi neutral at this moment of the episode.

The aim of this last paragraph is to estimate the impact of the heatwave on the dust net radiative effect. For that, two parameters directly related to the heatwave and needed for the computation of the LW radiative effect are considered: the surface temperature and the profile of the air temperature. In the following we calculate Dc $DRE_{LW}$ (the variations produced

on Df $DRE_{LW}$ are neglected since Df $DRE_{LW}$ is small) in a controlled way assuming the following three scenarios:

1. Heatwave-1 (HW1): This study with a climatological LST from the 8 years 2011-2018.
2. Heatwave-2 (HW2): This study with an air temperature profile in the dust layer equal to the actual one minus 6 ºC.
3. Heatwave-3 (HW3): This study with a climatological LST from the 8 years 2011-2018, and an air temperature profile in the dust layer equal to the actual one minus 6 ºC (equivalent to a combination of HW1+HW2 scenarios).

The climatological LST was calculated as the average over the 8 years preceding 2019 (2011-2018). The data were provided by the Copernicus Global Land Service in the way than the hourly LST of 23 – 30 June, 2019 (see Section 2.3). Although the analysis was originally planned for a 10-year climatology, Copernicus data of the LST are available only back to 2011. In HW1 and HW3 the LST has been replaced by the 8-year climatological value. According to AEMET (2019), the temperature anomaly at 850 hPa on 28 June, 2019, in the region of Barcelona was on the order or +12 ºC. This is probably the maximum

of the episode (see Figure 6 and Figure 13c). As a rule of thumb the mean anomaly over the episode is estimated to be +6 ºC. For comparison the difference between the maximum daily LST (32.7ºC reached on 28 June, Figure 6) and the mean over the episode (28.8ºC) is 3.9ºC. In HW2 and HW3 6ºC have been subtracted to the temperature profiles in the dust layer. The different scenarios allow to estimate how the dust radiative effect would have been modified: HW1, if the surface temperature had not been so high; HW2, if the air had not been so warm; and HW3, if the dust episode had not been accompanied by a

heatwave ("normal" surface and air temperature). Figure 14a and b show the daily DD $DRE_{NET}$ and the daily Dc LW/SW DRE ratio at SRF and TOA, respectively, for the three scenarios as well as in the heatwave conditions (this study). Figure 14c shows the daily LST (2019) and the climatological LST (2011-2018) for comparison. From the latter figure, one sees that at the beginning of the episode (23 and 24 June) the 2019 daily LST is indeed lower than the climatological one. From 24 June onwards it increases strongly. On the last three days of the episode (28, 29 and 30 June) the 2019 daily LST is approximately

4.0 – 4.5 ºC above the climatological LST. In HW1 (red lines in Figure 14a and b) the use of the climatological LST has no impact on the Dc LW/SW DRE ratio at SRF (and thus on the DD $DRE_{NET}$ either). Contrarily the impact at the TOA is quite noticeable: towards the end of the episode, the reduction of LST in HW1 yields logically a reduction in Dc $DRE_{LW}$, thus a reduction in the Dc LW/SW DRE ratio and thus an amplification of the dust cooling effect at TOA. Say the other around, the effect of a high LST during the heatwave is to increase the amount of radiation which escapes to space. Since the radiation

budget at SRF is nearly unchanged, the effect in the atmosphere is to reduce the heating of the atmosphere. In HW2 (blue lines in Figure 14a and b) the use of a decreased air temperature in the dust layer has an impact at both SRF and TOA and of opposite sign. The decrease of the air temperature has a direct impact on the gaseous transmittance. According to Dubuisson et al. (2004) and following the correlated k-distribution (Lacis and Oinas, 1991) used in GAME, a decrease in air temperature yields a decrease in the absorption coefficient. This will result in more LW radiation propagating upward in the atmosphere and thus

in a reduction of the dust Dc $DRE_{LW}$ in the dust layers and consequently at SRF (Figure 14a), and in an increase of the LW radiation reaching TOA (Figure 14b). Consequently, when the air temperature in the dust layer is reduce by 6 ºC, DD $DRE_{NET}$ is shifted towards negative values at SRF and towards positive values at TOA. Seen from the heatwave perspective, the effect of high temperatures in the dust layer during the heatwave is to reduce the dust net cooling effect at SRF and to amplify it at





TOA (decrease of the amount of radiation which escapes to space). These opposite variations will have the effect of reducing
the atmospheric heating provoked by the dust particles. Finally, when both HW1 and HW2 scenarios are considered
simultaneously (HW3), their effect on the daily DD $DRE_{NET}$ and the daily Dc LW/SW DRE ratio combine (green lines in
Figure 14a and b). At SRF DD $DRE_{NET}$ is shifted towards negative values like in HW2 (in HW1 DD $DRE_{NET}$ was nearly
unchanged). At TOA HW1 and HW2 have opposite effects on DD $DRE_{NET}$ in the second half of the episode when the 2019
LST is larger than the climatological LST; consequently their effects compensate and DD $DRE_{NET}$ in HW3 is practically
unchanged with respect to the results of this study. Interestingly, in the first half of the episode when the 2019 LST is smaller
than the climatological LST, both HW1 and HW2 produce a shift of DD $DRE_{NET}$ at TOA towards positive values which result
in a reduction of the dust net cooling effect in HW3. From the heatwave perspective, we conclude that the higher LST and the
higher air temperature due to the heatwave that accompanied the dust episode 1) provoked a reduction of the dust net cooling
effect at the surface, 2) left unchanged the dust net cooling effect at TOA, and 3) consequently reduced the dust net heating of
the atmosphere.

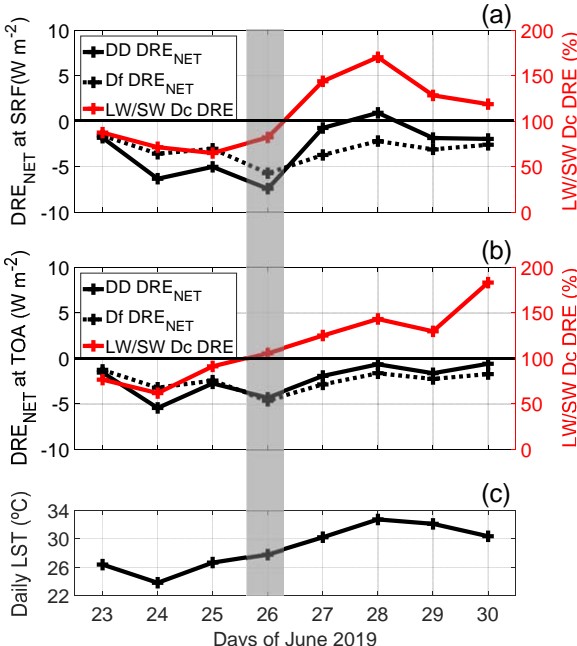

**Figure 13. (a) Daily Df and DD $DRE_{NET}$ (left axis) and Daily Dc LW/SW DRE ratio (right axis) at SRF; (b) Daily Df and DD
$DRE_{NET}$ (left axis) and Daily Dc LW/SW DRE ratio (right axis) at TOA; (c) Daily mean LST. Site: BCN; Time period: 23-30 June.
The shaded area represents the period during which both SRF and TOA Dc LW/SW DRE ratio cross the 100 % threshold.**

## 4 Conclusions

This companion paper of Córdoba-Jabonero et al. (2021) estimates the temporal variation of the instantaneous and daily dust
longwave and net direct radiative effect during an intense dust episode that occurred between 23 and 30 June, 2019, and
coincided with a mega-heatwave. It also investigates the effect of the heatwave on the dust radiative effect. The radiative effect
was calculated with the GAME radiative transfer model separately for the fine- and coarse-mode dust. The dust radiative
properties in the longwave spectral range were calculated with a Mie code and particle microphysics from AERONET. The
dust fine- and coarse-mode vertical distribution as well as the shortwave DRE were taken from the companion paper. Two






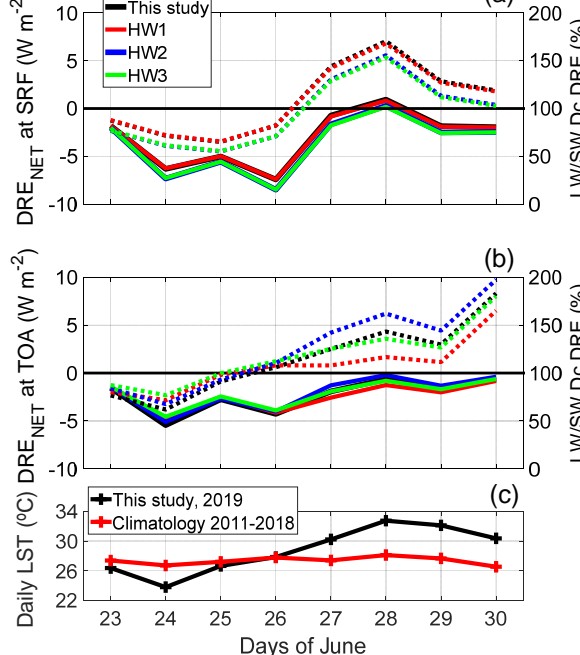

**Figure 14. Daily DD $DRE_{NET}$ (solid lines, left axis) and Daily Dc LW/SW DRE ratio (dotted lines, right axis) at (a) SRF and (b) TOA; (c) Daily mean LST. Site: BCN; Time period: 23-30 June. HW1: climatological LST; HW2: T(air)-6 ℃ in the dust layer; HW3: climatological LST and T(air)-6 ℃ in the dust layer. In the LST plot, the climatological surface temperature is also plotted.**

sites are considered: Barcelona, Spain, (23-30 June) and Leipzig, Germany, (29-30 June) to estimate the impact of the dust transport on its DRE.

Independently of the atmospheric level where it is estimated, the instantaneous Df $DRE_{LW}$ are low and do not exceed +0.6 W m$^{-2}$ in BCN and +0.2 W m$^{-2}$ in LPZ. The respective daily values do not exceed +0.42 and +0.08 W m$^{-2}$. In average over the whole event the contribution of Df daily $DRE_{LW}$ to the total dust $DRE_{LW}$ is not higher than 4 %. Most of the dust

$DRE_{LW}$ is thus produced by the coarse-mode particles. Instantaneous Dc $DRE_{LW}$ at SRF in BCN reaches a maximum of +11.4 W m$^{-2}$ and the daily values a maximum of +9.8 W m$^{-2}$. In LPZ the instantaneous and daily values reach maxima at +3.3 and +1.3 W m$^{-2}$, respectively. Dc $DRE_{LW}$ at TOA is smaller than at SRF which produces a negative (cooling) effect in the atmosphere. Df $DRE_{LW}$ in BCN represents 6.7 (3.7) % at SRF (TOA) of its SW counterpart. The Dc LW/SW DRE ratio in BCN increases from ~80 % at the beginning of the episode up to 170 % towards the end of it. Such an unusual tendency is

attributed to increasing coarse-mode size and surface temperature along the episode. In the last four days of the episode Dc $DRE_{LW}$ is larger than Dc $DRE_{SW}$ (Dc LW/SW DRE ratio > 100 %) at both SRF and TOA. This is a singular result of this study which has the effect of reducing considerably the SW cooling.

The results of the dust net DRE are discussed in terms of daily values which are the magnitude that matters for assessing the effect of aerosols on the Earth-Atmospheric radiative budget. The fine mode $DRE_{NET}$ is similar to $DRE_{SW}$ since

$DRE_{LW} \ll DRE_{SW}$. In BCN Dc $DRE_{NET}$ at SRF is negative at the beginning of the event and positive afterwards (peak at +3.1 W m$^{-2}$). In LPZ Dc $DRE_{NET}$ is slightly negative (mean of -0.25 W m$^{-2}$). At TOA the same tendency is observed in BCN (peak at +3.1 W m$^{-2}$). But in LPZ Dc $DRE_{NET}$ is also positive at TOA (mean of +0.50 W m$^{-2}$). This stronger effect of the dust $DRE_{LW}$ at TOA (vs. SRF) in LPZ is related with the quite high dust plume in LPZ (> 3.3 km). Overall, the total dust net DRE is small: -3.0 (-2.3) W m$^{-2}$ at SRF (TOA) in BCN and -1.40 (-0.35) W m$^{-2}$ in LPZ. On one occasion at SRF in BCN DD





$DRE_{NET}$ is even positive (+0.9 W m$^{-2}$ on 28 June) indicating a total dust net warming at the surface, which contrasts with the "traditional" dust cooling effect usually observed in clear sky conditions.

As far as dust direct radiative efficiency is concerned, Df LW DREff is small so that Df net DREff is nearly equal to Df SW DREff. Dc LW DREff is positive which contributes to decrease significantly the net DREff with respect to the SW one. Overall, the total dust net DREff is: -54.1 (-37.3) W m$^{-2}$ $\tau^{-1}$ at SRF (TOA) in BCN and -85.3 (-114.2) W m$^{-2}$ $\tau^{-1}$ in LPZ. The

higher dust direct radiative efficiency in LPZ (vs. BCN) is due to the SW component, already discussed in the companion paper. At TOA the difference between of DD net DREff between both sites is caused by the larger compensation of the LW component in BCN (vs. LPZ). Finally it should be noted that, in BCN at the event scale and for the total dust, the inclusion of the LW DRE in the calculation of the total net DREff reduces a factor 1.6 the SW DREff at SRF and BOA and a factor 1.8 in the atmosphere. These reduction factors significantly larger than 1 highlight the importance of considering the longwave

component in studies focused on mineral dust radiative effects.

In order to evaluate the impact of the heatwave that accompanied the dust intrusion on the dust radiative effect, we perform a sensitivity study on the surface temperature and the air temperature in the dust layer, both linked to the heatwave and upon which the LW DRE strongly depends. Three scenarios are considered: 1) the LST is set to climatological values, 2) the air temperature in the dust layer is reduced of 6 ºC, and 3) a combination of the first two scenarios. Our findings show that

the increase of LST and air temperature in the dust layer caused by the heatwave 1) provoked a reduction of the dust net cooling effect at the surface, 2) left unchanged the dust net cooling effect at TOA, and 3) consequently reduced the dust net heating of the atmosphere. The situation at the surface is a vicious circle: the heatwave reduces the dust cooling effect which, in turn, may increase some critical variables associated to the heatwave (e.g., LST and air temperature). The effect of the heatwave on the dust radiative effect is reverse as it contributes to cool down the atmosphere. Since recent studies have warned

that mega-heatwaves such as the one studied in this work might become more frequent in the future, the novel results presented in this paper call for more research on the subject.

**Data availability.** Part of the data used in this publication were obtained as part of the AERONET and MPLNET networks and are publicly available. For additional data or information please contact the authors.

**Author Contributions.** MS and CC-J designed the study and wrote the original draft paper. CC-J, MS and AA provided data.
CC-J, MS and MALC performed data analysis with contributions from AA, AC, M-PZ, AR-G- and CM-P. All authors reviewed and edited the final version of the manuscript. All the authors agreed to the final version of the paper.

**Competing interests.** The authors declare that they have no conflict of interest.

**Acknowledgments.** The MPLNET project is funded by the NASA Radiation Sciences Program and Earth Observing System. The MPLNET staff at NASA GSFC is warmly acknowledged for the continuous help in keeping the P-MPL systems and the
data analysis up to date. We particularly thank E.J. Welton for providing the P-MPL unit in place at the Barcelona site.

**Financial support.** This research was funded by the Spanish Ministry of Science, Innovation and Universities (CGL2017-90884-REDT and PRX18/00137 "Salvador de Madariaga" programme), the Spanish Ministry of Science and Innovation (PID2019-104205GB-C21 and PID2019-103886RB-I00), the H2020 programme from the European Union (GA no. 654109, 778349, 871115 and 101008004), and both Units of Excellence "María de Maeztu" (MDM-2016-0600 and MDM-2017-0737)
financed by the Spanish State Research Agency (AEI). MALC is supported by the INTA predoctoral contract programme.



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
