# Peer review of "Aerosol radiative impact during the summer 2019 heatwave produced partly by an inter-continental Saharan dust outbreak – Part 2: Longwave and net dust direct radiative effect"

_Atmospheric Chemistry and Physics, 2021_

## Referee Comment (RC2)

[referee-annotated manuscript omitted]

---

## Author Comment (AC1)

The presented paper is the companion paper of Córdoba-Jabonero et al. (2021). It is focused on the characterization of atmospheric dust properties and on the net and longwave radiative effect of that dust during a mega-heatwave at Barcelona (23 to 30 June, 2019) and at Leipzig (29 to 30 June, 2019). Authors use polarized multi-pulse lidar measurements and AERONET products to determine the microphysical dust properties and a Mie code to obtain the radiative properties in the longwave range. GAME radiative transfer model is used to study the radiative effect of the detected coarse and fine dust. They also test the impact of the extreme local surface temperature and the temperature of the dust layer on the dust radiative effect. This paper fits with the scope of the journal and the obtained results are interesting for the scientific community, hence, I recommend its publication after minor corrections.

**Reply**: Thank you very much. We greatly appreciate the reviewer positive feedback.

Minor comments:

P2L59: "classified as mega-heatwave by some authors", please, cite these authors.

**Reply**: The following information has been added: "classified as mega-heatwave by Sousa et al. (2019) and Xu et al. (2021)".

P3L89: "OLR of -0.8 % and a root mean square error of 2.52 W m-2."  Could be added the RMSE in %, or the bias in Wm-2?

**Reply**: The more logical here would be to express the bias in $Wm^{-2}$ as it is its original mathematical definition. This is what we have done here with the data from the paper of Sicard et al. (2014) that, luckily, we were able to find. We find a bias of -0.82 $Wm^{-2}$, which in relative terms is -0.78 % (-0.8 % is the value rounded up in the original paper). In the revised paper we now give a bias of -0.8 $Wm^{-2}$.

P3L96: Why LW is calculated only between 5 and 19? The Earth is emitting LW also at night-time. That is the weakest point of this work. It is mentioned that GAME cannot be used to calculate radiation fluxes at night-time (P8L223), but the contribution of dust to radiative forcing at night-time is also important. It is not clear how the daily DRE_LW and DRE_SW values are calculated. The manuscript seems to indicate that these daily averages are obtained by averaging all the DRE values available in a day, that is, the values obtained only during the hours of sunshine (daytime). The DRE_SW values at night-time (values equal to 0) should be included in the calculation of the average of the daily DRE_SW. In addition, the daily DRE_LW averages should be also calculated taking into account night-time data. The authors should clarify how they calculate the DRE daily averages and try to add night-time values if they are not included.

**Reply**: Referee #2 made a similar comment. Thank you. First we have deleted in Section 2 the reference to the time interval (5-19 UTC) so that the full explanation is given in the first paragraph of Section 3. This first paragraph was entirely re-written as follows to explain how SW and LW daily values were computed:

"Instantaneous and daily averages are presented. Instantaneous values are calculated from 5 to 19 UTC because GAME calculates shortwave and longwave fluxes only during the hours of sunshine (daytime) as it computes both components of the sun. Daily averages in the shortwave are calculated as the mean (over 24 hours) of the 15 instantaneous daytime values. The assumption is made here that shortwave fluxes during nighttime are zero. Daily averages in the longwave are more difficult to estimate because longwave fluxes during nighttime are not zero unlike in the shortwave. They were calculated as the mean (over 15 hours) of the 15 instantaneous daytime values. The assumption behind this calculation is that the nighttime longwave fluxes are equal to the mean of the daytime fluxes. A similar assumption has been made by Di Sarra et al. (2011) who averaged instantaneous $DRE_{LW}$ obtained every 6 hours and by Meloni et al. (2015) who assumed their instantaneous $DRE_{LW}$ retrieval constant throughout the 24 hours."

Discussion about Figure 2: The fine radius is reduced with time while the coarse one increases. Authors mention the "aerosol aging" and the "Secondary aerosol formation is enhanced in stagnant" as possible phenomena behind this behaviour. The geometric median radius and standard deviation were assumed equal to the retrieved by AERONET. These retrieved values are for all aerosol in the atmospheric column, not only dust, hence the presence of other aerosols could also modify the fine and coarse radius that are assumed for dust in this work. Changes on the other aerosols could be partially responsible of the observed changes in fine and coarse radius. Authors should take this issue also as a possible source of uncertainty, since the obtained results could be affected by the variation caused by these other aerosols (the retrieved radius and standard deviations at the column should not exactly be the same for the dust). Regards the uncertainty on the aerosol properties and the obtained radiative ones, could be helpful to quantify how they are propagated in the radiative transfer simulations in order to provide a confidence interval in the obtained DRE results.

**Reply**: Interestingly Referee #2 made again a similar comment.

It is worth recording that the typical aerosol background in Barcelona is made of fine-mode particles, and thus the coarse mode observed during the dust outbreak is fully attributed to the presence of mineral dust. So it appears legitimate to take the AERONET columnar properties of the coarse mode as representative of the coarse mode mineral dust only (without mixing with other aerosols). We also recall that the application of POLIPHON (that was described in the companion paper) quantifies the contribution of the dust coarse mode from the fine mode (dust + other aerosols) on the basis of their depolarizing properties.

The changes in the fine mode could be due to changes in the fine-mode dust, fine-mode non-dust or both components, as the referee says. However this paper part II presents the longwave radiative effect and the results show that the fine-to-total ratio of LW DRE is not larger than 5 %, i.e. that fine-mode particles (dust or/and other aerosols) contribute very little to the LW DRE. We conclude that changes in the fine mode radius and sigma do not represent a relevant source of uncertainty.

P7L204: Why surface albedo at Leipzig is assumed equal to Barcelona albedo? Is it possible to estimate the climatological value of albedo at Leipzig as in Barcelona?

**Reply**: We thank the referee for the suggestion. It is certainly a point to verify in the future. This estimation of the LW albedo in Leipzig, although not properly difficult, is laborious and time-consuming. CERES data have to be requested and the delivery time is quite random. Once we have the data we need to apply spatial filters to select only the pixels representative of the site of interest and then apply further criteria to remove possible outliers. At this stage of the paper we prefer to rely on the climatological analysis already done in Barcelona. And there are at least two reasons for that:

1. In Sicard et al. (2014a, see the references in the paper) from where the mean value (0.017) is taken, the standard deviation is also given. It is 0.001 which suggests that the variability of this parameter is very small and that the LW surface albedo in a urban environment like Barcelona is stable with time. On this basis we assumed that the LW surface albedo in Leipzig should be similar to that of Barcelona.
2. In the following figure we show the global LW surface emissivity ($\varepsilon = 1 - \rho$) at 11.24 μm for the month of December from climatological data from IASI. It shows that in Europe the emissivity has no marked geographical variations (the orange color dominates) and is around 0.985 (albedo ~0.015) which is in agreement with the mean value retrieved from CERES in Barcelona. This second point gives credits to our hypothesis that Leipzig and Barcelona should have very similar LW surface albedo. The paragraph about the surface albedo has been completed as follows: "The same value was used at both sites and the explanation behind this is that the longwave surface albedo is quite stable over the whole European continent as shown by Zhou et al. (2013).".

[Figure]

This figure represents IASI December nighttime monthly mean climatology of the surface emissivity (i.e. 1-surface albedo) at 890 cm$^{-1}$, i.e. 11.24 μm. It has been taken from D. K. Zhou, A. M. Larar and X. Liu, "MetOp-A/IASI Observed Continental Thermal IR Emissivity Variations," in IEEE Journal of Selected Topics in Applied Earth Observations and Remote Sensing, vol. 6, no. 3, pp. 1156-1162, June 2013, doi: 10.1109/JSTARS.2013.2238892.

Figure 10. Y axis must be changed by the represented magnitude with its units.

**Reply**: Done.

P14L398: Please, remove "!".

**Reply**: Deleted.

P18L515: DRE_LW << DRE_SW, both terms should be marked as absolute values.

**Reply**: Done.

Section 3.2. Authors study the effect of the heatwave (high temperature at surface and at dust layer) on the DRE. But DRE also affects to the heatwave analysed parameters (surface and atmosphere temperatures), which affects to DRE, and changes on DRE affects again to temperature and again and again... Could be discussed if there are any positive or negative feedback process between DRE and heatwave? It is interesting to know if the intrusion of dust during a heatwave helps to mitigate this heatwave or on the contrary it enhances heatwave temperatures.

**Reply**: To do what the referee suggests, i.e. to study the interactions between aerosols, radiation and meteorology, a coupled model would be needed. And this work is out of the scope of our paper. In the conclusion of the paper some piece of information related to this comment is given: "The situation at the surface is a vicious circle: the heatwave reduces the dust cooling effect which, in turn, may increase some critical variables associated to the heatwave (e.g., LST and air temperature).". In summary at the surface the HW (in particular enhanced LST and air temperature in the dust layer) reduces cooling which may lead to increased LST and air temperature in the dust layer which may reduce further the dust cooling, etc. At the TOA, the HW has no significant effect on the dust net cooling.

It is interesting to note that Referee #2 made a similar comment suggesting to make the sensitivity study the other way around, i.e. looking at the effect of the dust outbreak on the heatwave. We copy our answer: "It is clear that the study the other way around would also have an interest. However as the primary focus of the paper is on the dust direct radiative effect, the sensitivity study is made to analyze the effect of the heatwave on the dust direct radiative effect.

The other way around would require to be able to compute an aerosol radiative effect caused by the heatwave first and a background aerosol load, and then add the mineral dust. At first sight it is conceptually a little more difficult to conceive."

---

## Author Comment (AC2)

Review of "Aerosol radiative impact during the summer 2019 heatwave produced partly by an inter-continental Saharan dust outbreak – Part 2: Long-wave and net dust direct radiative effect" by M. Sicard et al., 2021. The submitted manuscript, companion of ACP-21-6455, assesses, in radiative terms, the impact of a dust outbreak paired with an abnormal heatwave. The manuscript is scientifically interesting because those events will be more and more frequent because of climate change. The following issues should be addressed before publication:

**Reply**: Thank you very much. We greatly appreciate the reviewer positive feedback.

1) From MPLNET data, it looks like that the dust contribution drops drastically after 27 June 2019 at noon. How can the authors be sure that this is dust and not a mixture of dust and local aerosols? How this impacts the refractive index choice used as input in Mie code? This is a source of possible uncertainty.

**Reply**: On 27 June the dust outbreak intensity decreases but the changes are not drastic. The MPL data of both BCN and LPZ are indeed represented in Figure 5 of the companion paper in terms of volume depolarization ratio (VDR). According to the color bar it is true that the VDR seems smaller after 27 June, which reflects that the dust is not as pure as before 27 June. However, the application of POLIPHON takes this aspect into account, and the lower the VDR, the lower the concentration of the dust coarse mode. In these conditions the fine mode retrieval is representative of a mixing (fine background or local + fine dust). Since the fine-to-total ratio of LW DRE is not larger than 5 %, this aspect does not represent a relevant source of uncertainty.

2) The longwave contribution is computed just from 5 to 19 UTC. Is there any reason for this time interval? Longwave radiation is not depending on sunlight.

**Reply**: The code computes the LW radiation from the sun, among other radiation sources, and thus can only be run during the sun hours, i.e. from 5 to 19 UTC during summertime at the longitudes of Central European Time and mid-latitudes. In the revised manuscript in the last paragraph before section 3.1 we have added this explanation "as it computes the sun LW component".

3) The strength of this paper is paring the dust outbreak with a heatwave in terms of radiative effects. This should be better highlighted.

**Reply**: We have made a general effort to highlight this especially in the abstract, introduction and conclusion.

4) English should be revised because some sentences are not clear.

**Reply**: The sentences highlighted in the attached file have been reformulated and/or completed, when possible. We have also gone through a full reading and spelling revision of the paper.

Specific comments can be found in the attached file.

**Reply**: All comments have been addressed in the revised manuscript, and are marked in the version with "Change control" activated. Below we are giving an answer to the comments which require an answer:

RC2, page 1: this expression is too colloquial. I would say "coincident with one of the strongest heatwaves"
**Reply**: The prefix "mega" is not colloquial. Numerically it quantifies a unit being multiplied by $10^6$. It has been applied peer-review articles such as Barriopedro et al. (2011) or Xu et al. (2020b); see the paper. Barriopedro et al. (2011) give this definition of a mega heatwave:

The concept of mega-heatwave is herein used to refer to regional mean temperature anomalies (over ~1 million $km^2$) of extraordinary amplitude (approximately ≥3 SDs relative to the 1970–1999 period) at subseasonal scales (of at least 7 days), thus differing from the classic local heatwave definition.

In the recent years the prefix has been misused commonly in a colloquial way as a superlative. This may be the reason why it sounds too colloquial to the referee, but in its original meaning the prefix mega is not colloquial.

RC2, page 3: why exactly from 5 to 19? Longwave can be computed all over a day

RC2, page 8: why not on the whole day?

**Reply**: See answer to Comment #2.

RC2, page 3: Notation should be uniform. Before cm-1 was used.

**Reply**: Before (line 79-80 of the same page) we give the spectral limits in cm-1 because it is the way they are defined in the code but we also convert them to longwave to the readers who are more familiar with wavelength than wavenumbers. The referee will note that the spectral limits in wavenumber are only given in the Section 2 about the description of the RTM.

RC2, page 11: I suggest to add a plot for BCN with DD in x-axis s and NET DD in y-axis

**Reply**: We would like to know the idea of the referee for making this suggestion, and also whether he/she suggests to plot NET DRE vs. DOD or vs. LW DRE of the total dust (DD). It is not clear from the comment alone. For the time being, since the paper already includes 13 figures, and since we think this could be out of the scope of the paper, such a plot has not been included in the revised manuscript.

RC2, page 17: it would be much more interesting the other way-round, the effects of dust outbreak on the heatwave

**Reply:** It is clear that the study the other way around would also have an interest. However as the primary focus of the paper is on the dust direct radiative effect, the sensitivity study is made to analyze the effect of the heatwave on the dust direct radiative effect.

The other way around would require to be able to compute an aerosol radiative effect caused by the heatwave first and a background aerosol load, and then add the mineral dust. At first sight it is conceptually a little more difficult to conceive.